# Comprehensive Steroid Assay with Non-Targeted Analysis Using Liquid Chromatography Ion Mobility Mass Spectrometry

**DOI:** 10.3390/ijms232213858

**Published:** 2022-11-10

**Authors:** Mai Yamakawa, Shigehiro Karashima, Riko Takata, Taichi Haba, Keigo Kuroiwa, Hideaki Touyama, Atsushi Hashimoto, Seigo Konishi, Daisuke Aono, Mitsuhiro Kometani, Hidetaka Nambo, Takashi Yoneda, Issey Osaka

**Affiliations:** 1Department of Biotechnology and Pharmaceutical Engineering, Graduate School of Engineering, Toyama Prefectural University, Imizu 939-0398, Japan; 2Institute of Liberal Arts and Science, Kanazawa University, Kanazawa 920-1192, Japan; 3Department of Pharmaceutical Engineering, Faculty of Engineering, Toyama Prefectural University, Imizu 939-0398, Japan; 4Department of Electronics and Information Engineering, Graduate School of Engineering, Toyama Prefectural University, Imizu 939-0398, Japan; 5Department of Electronics and Information Engineering, Faculty of Engineering, Toyama Prefectural University, Imizu 939-0398, Japan; 6Department of Information Systems Engineering, Faculty of Engineering, Toyama Prefectural University, Imizu 939-0398, Japan; 7Faculty of Transdisciplinary Sciences, Institute of Transdisciplinary Sciences, Kanazawa University, Kanazawa 920-1192, Japan; 8Department of Endocrinology and Metabolism, Graduate School of Medicine, Kanazawa University, Kanazawa 920-8641, Japan; 9School of Electrical Information Communication Engineering, College of Science and Engineering, Kanazawa University, Kanazawa 920-1192, Japan; 10Department of Health Promotion and Medicine of the Future, Graduate School of Medicine, Kanazawa University, Kanazawa 920-8541, Japan

**Keywords:** steroid, aldosterone, quantitation, non-target, ion mobility, LC/IM/MS, aldosterone-related disorder

## Abstract

Aldosterone-producing adenomas (APAs) have different steroid profiles in serum, depending on the causative genetic mutation. Ion mobility is a separation technique for gas-phase ions based on their *m*/*z* values, shapes, and sizes. Human serum (100 µL) was purified by liquid–liquid extraction using *tert*-butyl methyl ether/ethyl acetate at 1/1 (*v*/*v*) and mixed with deuterium-labeled steroids as the internal standard. The separated supernatant was dried, re-dissolved in water containing 20% methanol, and injected into a liquid chromatography–ion mobility–mass spectrometer (LC/IM/MS). We established a highly sensitive assay system by separating 20 steroids based on their retention time, *m*/*z* value, and drift time. Twenty steroids were measured in the serum of patients with primary aldosteronism, essential hypertension, and healthy subjects and were clearly classified using principal component analysis. This method was also able to detect phosphatidylcholine and phosphatidylethanolamine, which were not targeted. LC/IM/MS has a high selectivity for known compounds and has the potential to provide information on unknown compounds. This analytical method has the potential to elucidate the pathogenesis of APA and identify unknown steroids that could serve as biomarkers for APA with different genetic mutations.

## 1. Introduction

Primary aldosteronism (PA) is a common secondary hypertension [1,2]. Compared to essential hypertension (EHT), PA is associated with aldosterone overproduction and more frequent cardiovascular complications [3,4]. *KCNJ5*, *CACNA1D*, *ATP1A1*, and *ATP2B3* have been reported as causative genes for aldosterone-producing adenomas (APAs) [5,6,7,8], and the molecular mechanisms of APA have been elucidated. Although these genetic abnormalities affect intracellular ion homeostasis and cause aldosterone overproduction, there are still some APAs with unknown causative genes.

In addition, the steroid profile of APA has been shown to differ depending on the causative genetic mutation [9]. William et al. reported that APAs with *KCNJ5* mutations have higher blood levels of the hybrid steroid 18-hydroxycortisol than others [9]. Experiments in cultured adrenal cells also suggest that cell lines transfected with the *KCNJ5 T158A* gene have increased levels of aldosterone, 18-hydroxycortisol, and 18-oxocortisol [10]. APAs with genetic abnormalities other than that of *KCNJ5* or with unidentified causes are expected to reveal new molecular biological mechanisms, as new steroid hormones other than hybrid steroids, such as 18-oxocortisol, may serve as biomarkers.

Steroid hormone measurements using immune assays have problems with accuracy because of cross-reactivity [11]. Therefore, mass spectrometry (MS)-based methods are preferred [11,12]. Liquid chromatography–mass spectrometry (LC/MS) and liquid chromatography–tandem mass spectrometry (LC/MS/MS) are the most common analytical methods for the quantitative analysis of known steroids, with no cross-reactivity. LC/MS is capable of non-targeted analysis; however, it has lower qualitative performance. LC/MS/MS is highly selective and sensitive; however, it is difficult to simultaneously perform non-targeted analysis for unknown compounds.

Ion mobility (IM) is a separation technique that relies on the size, shape, and charge ratio of ions [13]. Liquid chromatography–ion mobility–mass spectrometry (LC/IM/MS) is highly selective for known compounds and has the potential to provide information about unknown compounds. Currently, only a few qualitative and quantitative methods for steroids using IM have been reported. Hernandez-Mesa et al. evaluated the collision cross section (CCS) of steroid ions obtained by IM/MS to identify the steroid isomers [14]. Arthur et al. quantitatively analyzed various anabolic steroid metabolites in urine using non-target LC/IM/MS analysis [15]. If simpler quantitative and non-targeted analyses can be simultaneously performed, new molecular biological mechanisms of APA can be revealed from the qualitative and quantitative analyses of unknown steroid hormones whose functions and pathways are unknown. Our objective was to establish a method for the simultaneous quantification of minimal amounts of classical steroids in serum and non-targeted analysis of other compounds using LC/IM/MS.

## 2. Results

### 2.1. LC/IM/MS Measurement of Standard Solution Containing Steroids

For LC/IM/MS analysis, injection of 20 μL of the sample solution prepared with H_2_O/MeOH at 8/2 (*v*/*v*) allowed for the sensitive detection of steroids. Highly hydrophilic steroids, such as aldosterone, 18-hydroxycorticosterone, and 19-hydroxy-4-androstene-3,17-dione were adequately separated by adopting a gentle slope as a gradient condition for LC. Highly hydrophobic steroids, such as progesterone, testosterone, and androstenedione, can be separated easily with a steeper gradient program. The peak shape of compounds with a long retention time (RT) at a constant flow rate of the mobile phase is broad, generally because of the slow flow speed in LC. A gradient condition with a combination of gentle and steep slopes was adopted to obtain sharp peaks and high sensitivity.

Standard and serum samples were analyzed using LC/IM/MS. The obtained mass-to-charge number ratio (*m*/*z*) values, RT, and drift time (DT) of detected steroids are summarized in Table 1. The largest error in the *m*/*z* values was 0.003, with high accuracy. The extracted ion chromatograms of 20 steroid standard solutions with each specific DT at a concentration of 10 ng/dL are shown in Figure 1. In the mass chromatogram obtained via LC/IM/MS, 18-hydroxycorticosterone, aldosterone, 19-hydroxy-4-androstene-3,17-dione, testosterone, androstenedione, and progesterone were observed at 14.4, 14.6, 16.6, 26.0, 26.8, and 31.8 min, respectively. Although cortisol was not separated from cortisone via LC, it could be distinguished by the difference in *m*/*z* values. The DT has two axis labels: bins and millisecond (ms). In this instrumental condition the labels have the following relationship:(1)[DT (ms)]=0.039×[DT (bins)]−0.039

Steroid peaks were observed in an ion mobilogram obtained via LC/IM/MS. Two-dimensional plots of DT-RT integrating the ion mobilogram and mass chromatogram at RT 10–33 min, DT 38–65 bins, and *m*/*z* 285–380 are shown in Figure 2. Spots for each steroid were observed in the 2D plots. Testosterone (*m*/*z* 289, RT 26.0 min, and DT 43 bins), androstenedione (*m*/*z* 287, RT 26.8 min, and DT 43 bins), deoxycorticosterone (*m*/*z* 331, RT 26.5 min, and DT 50 bins), 17α-hydroxyprogesterone (*m*/*z* 331, RT 27.5 min, and DT 53 bins), 18-hydroxy-11-deoxy corticosterone (*m*/*z* 347, RT 20.6 min, and DT 54), 21-deoxycortisol (*m*/*z* 347, RT 22.0 min, and DT 52 bins), corticosterone (*m*/*z* 347, RT 22.3 min, and DT 54 bins), and 11-deoxycortisol (*m*/*z* 347, RT 22.9 min, and DT 55 bins) were separated using LC. Aldosterone and 18-hydroxycorticosterone, with a difference of only 2 Da, were clearly separated using IM. Isomers (*m*/*z* 347), 21-deoxycortisol (RT 22.0 min, DT 52 bins), and 11-deoxycortisol (RT 22.9 min, DT 55 bins) were separated using LC and IM. From the above results, steroids were separated and detected using LC/IM/MS, and spots of these steroids were observed in 2D plots.

Calibration curves were created for each of the 20 steroids, and they all yielded good linearities, as shown in Appendix A. The coefficient of determination (R2) for steroids was ≥0.994.

### 2.2. LC/IM/MS Measurement of Human Serum

Human serum spiked with a standard steroid mixture and an internal standard was analyzed using LC/IM/MS. The recovery rates of steroids in the serum were determined. A summary of the limit of detection (LOD), the limit of quantitation (LOQ), analytical recoveries, and matrix effect for LC/IM/MS of serum samples are shown in Table 2. The quality control (QC) values of the recoveries were run in replicates (*n* = 5). Cortisone, cortisol, corticosterone, and testosterone were evaluated using a 100-fold diluted serum. Recovery rates for most steroids were greater than 80%. The LOD was determined at a concentration of ≥3 signal-to-noise (S/N) ratio. The LOQ was determined at concentrations ≥5 S/N ratio and <20% relative standard deviation (RSD). The analysis of a small amount of aldosterone as the final metabolite originating from cortisol required an LOD of 3 ng/dL because of the low aldosterone levels in healthy individuals. LODs were 1.0, 1.0, 1.0, and 0.5 ng/dL for aldosterone, 18-hydroxycorticosterone, testosterone, and androstenedione, respectively. This indicated that quantitative analysis of aldosterone and other steroids with high sensitivity could be performed using LC/IM/MS.

A marginal ion suppression was observed as the matrix effect derived from serum components (Table 2). Early-eluting compounds from LC have a lower matrix effect than late-eluting compounds do. The matrix effect coefficient of late-eluting compounds was not close to 1.0. However, we decided to use the obtained calibration curves for quantitative analysis because of the small matrix effect and high recovery rate. Androstenedione and progesterone with high matrix effect had low recovery rates.

Serum specimens from four individuals were analyzed using LC/IM/MS. These included two healthy subjects: one with EHT and the other with PA. The steroids detected are shown in Table 3. The aldosterone concentrations in the sera of patients with EHT and PA were higher than those in healthy subjects. The activity of *CYP11B2*, an aldosterone synthase, is expressed as 18-oxocortisol (18-oxoF)/cortisol ratio or aldosterone/corticosterone ratio. The 18-oxoF/cortisol ratio was 0.000361, 0.000187, 0.000426, and 0.000745 for healthy subject 1, healthy subject 2, EHT patients, and PA patients, respectively. The aldosterone/corticosterone ratio was 0.013571, 0.016875, 0.026667, and 0.065385, respectively. Compared with that in the other subjects, *CYP11B2* activity was the highest in patients with PA.

It is difficult to evaluate the correlation between all steroids. Principal component analysis (PCA) was used to statistically analyze the steroidal differences in each specimen. The results are shown in Figure 3. PCA score plots showed clear separation and grouping of healthy and hypertensive samples, respectively. The PCA loading plots show that steroid ions contribute to clustering. The X-axis and Y-axis describe the first and second principal components (PCs), respectively. The major contributors to the differences between healthy patients and other patients were aldosterone, 11-deoxycortisol, 18-hydroxycortisol, and 18-hydroxycorticosterone. The results suggest that PCA analysis with LC/IM/MS data has the potential to group patients with hypertension and discover novel biomarkers.

### 2.3. Non-Targeted Analysis Using LC/IM/MS

LC/IM/MS yielded unexpected lipid ions in steroid analysis owing to its non-targeted analysis with no MS/MS. It allows the detection of phosphatidylcholine (PC) and phosphatidylethanolamine (PE) of lipids in serum, with the simultaneous quantitation analysis of steroids. The extracted ion chromatograms of lipids are shown in Figure 4. Ions corresponding to *m*/*z* values of PE(36:3), PE(40:5), PE(40:4), PC(38:0), and PC(40:4) were observed. The longer alkyl chains and fewer double bonds of lipid molecules lead to a late RT in LC.

## 3. Discussion

We established a system for steroids that simultaneously enables targeted measurement and non-targeted analysis using LC/IM/MS. In addition to classical steroid hormones, 20 types of steroids, including hybrid steroids, such as 18-oxocortisol and 18-hydroxycortisol, can be measured with high sensitivity. Using this method, steroid profiles in the sera of the patient with PA, patient with EHT, and healthy subjects were measured and clearly classified using PCA. Quantitative analysis of these steroids simultaneously detected PC and PE, which were not targeted lipids. However, our method does not require derivatization and can identify non-targeted steroids and lipid metabolites, regardless of the ease of compound derivatization.

The comprehensive measurement of steroids using MS is a very useful tool in PA research. Adina et al. measured 17 steroids from adrenal venous sampling and peripheral serum and examined their usefulness for PA subtype classification. They showed that 11β-hydroxyandrostenedione and 11-deoxycortisol were superior to cortisol from the interpretation of adrenal venous sampling data and that a multi-steroid panel measured in peripheral blood was useful for the stratified classification of PA [16]. In addition, Eisenhofer et al. reported that comprehensive plasma steroid profiling combined with machine learning could facilitate screening for PA and identify patients with unilateral adenomas due to pathogenic *KCNJ5* variants [17]. The vast amount of information obtained from comprehensive steroid profiling in combination with artificial intelligence analysis, such as machine learning, may contribute to the understanding of the pathogenesis of aldosterone overproduction and provide more efficient and less invasive testing.

Ion mobility spectrometry (IMS) is a technique used to separate compounds in a mixture by CCSs. The IMS cell is filled with nitrogen gas (N_2_), and the electrostatic field applied in the direction of travel causes the ions to move. The bulkier compounds collide more frequently with N_2_ molecules, resulting in lower mobility. The difference in the resulting transit time (DT) allows for the separation of compounds. The CCS of an ionized compound depends only on the DT and *m*/*z* and is unique to that compound. Therefore, IMS-MS, which combines IMS with MS, enables the separation of components with the same molecular weight and the highly accurate analysis of the steric structure of compounds based on the compositional formula estimated by MS and the CCS information obtained by IMS. Furthermore, when used in combination with LC, IMS-MS is also effective for qualitative analysis of structural isomers that cannot be separated by LC [18,19,20,21]. The differences in the steric steroid ring structure related to structural isomers have markedly different biological effects in the human body [22,23]. Steroids are compounds with several structural isomers. Fourteen (e.g., *m*/*z* 361.200; cortisone and aldosterone) of the twenty steroids have structural isomer relationships in this study. These steroidal structural isomers were also clearly separated using three labels: RT, *m*/*z*, and DT. The principle of LC/IM/MS is well suited for the quantification of steroids, as structural isomers can be clearly separated. Hernandez-Mesa and Arthur separated steroid isomers by LC/IM/MS [14,15]. To improve the separation of isomers by IM, formed steroid alkali metal adduct ions [M + alkali metal]^+^ were used instead of proton adduct ions [M + H]^+^. The advantage of our LC/IM/MS method over LC/MS/MS, as a steroid determination method, is that it can simultaneously perform highly sensitive quantitative and non-targeted analyses.

Indeed, the analysis of protein complexes using IM/MS has been reported [24,25], and Baker et al. reported the application of LC/IM/MS for proteomics [26]. LC/IM/MS has also been used for metabolomic analysis [27], improving the S/N ratio of lipids by reducing chemical noise. For urinary steroid analysis, LC/IM/MS has been reported to separate steroids and their isomers for anti-doping analysis [28]. Velosa et al. evaluated the [M + H]^+^ and [M + Na]^+^ DTs of anabolic steroids using LC/IM/MS and calculated CCS values [29]. The results with multiple ionic species of [M + H]^+^ and [M + Na]^+^ enabled secondary characterization and improved the resolution of stereoisomers. However, [M + Na]^+^ is non-volatile and may not be suitable for continuous quantitative analysis because it is more likely to contaminate a mass spectrometer than [M + H]^+^. As described above, LC/IM/MS has the potential to improve the reliability of screening for known steroids and provide structural information for the elucidation of unknown compounds, even steroids. This may allow us to classify common endogenous steroids and synthetic/exogenous compounds more accurately or to discover new steroid metabolic pathways in vivo. LC/IM/MS may also be beneficial in the search for biomarkers and drug targets for the diagnosis and prediction of disease severity.

In clinical practice, steroid hormones have been measured using immunoassays (IAs) [30,31,32]. However, IAs are questionable for the analysis of complex biological samples, such as human specimens, because of their cross-reactivity. IAs are inferior to MS-based assays, which can measure multiple components simultaneously, because they can measure only one compound at a time. The ability to measure multiple steroids thoroughly at once, the lack of cross-reactivity, low LOD, and high accuracy are characteristics that make LC/IM/MS superior to IAs.

This study has several limitations. First, it measured only a small number of human serum samples. More cases need to be studied, particularly for evaluation using statistical analysis. Second, regarding the optimization of the pretreatment, there is a trade-off between contamination of the mass spectrometer and the detection sensitivity of non-targeted analysis [33]. For quantitative analysis, exact extraction is necessary to reduce the signal noise. In contrast, for non-targeted analysis, over-extraction results in fewer compounds being detected. Inadequate pretreatment is not suitable for continuous measurements because the sample contaminates the column and instrument and requires frequent cleaning. It is necessary to further investigate the optimal conditions for efficient non-targeted analysis with high sensitivity while minimizing cleaning operations associated with contamination. Finally, the absence of a compound library for LC/IM/MS precludes easy compound identification. The accumulation of RT, *m*/*z*, and DT data for compounds is needed in the future.

## 4. Materials and Methods

### 4.1. Materials

The analyzed steroids, androstenedione, testosterone, adrenosterone, 11-ketotestosterone, 19-hydroxy-4-androstene-3,17-dione, 11-deoxycorticosterone, 17α-hydroxyprogesterone, 21-deoxycortisol, aldosterone, cortisone, and 18-hydroxycorticosterone were purchased from Sigma-Aldrich (St. Louis, MO, USA). Corticosterone was purchased from Tokyo Chemical Industry Co. Ltd. (Tokyo, Japan). The 16α-hydroxyprogesterone, 18-oxocortisol, and 18-hydroxycortisol were purchased from Toronto Research Chemicals (Toronto, ON, Canada), and 11-deoxycortisol, 18-hydroxy-11-deoxycorticosterone, and progesterone were purchased from Makor Chemicals Ltd. (Jerusalem, Israel). The 11β-hydroxyprogesterone was purchased from ChemScene LLC (Monmouth Junction, NJ, USA). Cortisol was purchased from FUJIFILM Wako Pure Chemical Corporation (Osaka, Japan). The 11-ketoteststerone-[16,16,17-d3] and 18-hydroxycortisol-[9,11,12,14-d4] were purchased from IsoSciences (Ambler, PA, USA), and 21-deoxycortisol-d8 was purchased from Cerilliant-Certified Reference Materials (Round Rock, TX, USA). Internal standards of the AbsoluteIDQ^®^ Stero17 Kit were purchased from Biocrates Life Sciences AG (Innsbruck, Austria). For liquid–liquid extraction (LLE), tert-Butyl methyl ether (Kanto Chemical Co. Inc., Tokyo, Japan) and ethyl acetate (FUJIFILM Wako Pure Chemical Corporation, Osaka, Japan) were used. LC-MS grade acetonitrile and methanol were purchased from Kanto Chemical Co. Inc. Tetrahydrofuran (Kanto Chemical Co. Inc.) and ammonium fluoride (Sigma Aldrich) were used as the mobile phase for LC. Ultrapure water was obtained using a Milli-Q integral system (Merck Millipore, Burlington, MA, USA).

### 4.2. Sample Preparation

The mixed standard stock solution of 20 steroids and 3 deuterium-labeled steroids were prepared with water, methanol, and acetonitrile from a concentration of 0.1 mg/mL to 1 mg/mL. Thereafter, the non-labeled stock solution was diluted with methanol to 20,000, 6000, 2000, 600, 200, 60, 40, 20, and 10 ng/dL as steroid standard solutions. Seventeen deuterium-labeled steroid mixtures from the AbsoluteIDQ^®^ Stero17 Kit were dissolved in methanol. The 17 deuterium-labeled steroid solutions were diluted five times with three deuterium-labeled steroid solutions, methanol, and water. The three deuterium-labeled steroids were prepared at 60 ng/mL. The obtained 20 deuterium-labeled steroid solution in methanol/water at 8/2 (*v*/*v*) was used as the internal standard solution. These solutions were stored at −24 °C in glass tubes.

The sera were stored at −80 °C in plastic tubes. Serum samples from a healthy person were purchased from Cosmo Bio Co. Ltd. (Tokyo, Japan) as healthy 1. Healthy adult serum samples were obtained from a 34-year-old male volunteer as healthy 2. There were two patients: one was a 40-year-old man with PA who was administered spironolactone, and the other was a 45-year-old man with EHT who was administered angiotensin receptor blockers and calcium channel blockers. Written informed consent was obtained from all three subjects. Sera containing 1, 2, 3, 10, 30, 100, and 300 ng/dL steroids were prepared by the addition of 5 µL methanol solution of steroid standard. The solution was mixed with 5 µL internal standard solution in a 1.5 mL tube. These solutions were used for creation of calibration curves. The 100-fold water-diluted serum samples were prepared for analysis of cortisone, cortisol, corticosterone, and testosterone. Serum steroids were purified using LLE with *tert*-butyl methyl ether (MTBE)/ethyl acetate at 1/1 (*v*/*v*). LLE is a commonly used steroid extraction method [10,34]. We have improved the protocol for non-target LC/IM/MS. MTBE/ethyl acetate solution (400 µL) was added to the serum sample. The obtained solution was vortexed (VTX-3000L, LMS Co. Ltd., Tokyo, Japan), centrifuged (800× *g*, 1 min, KUBOTA Model 3520, Kubota, Tokyo, Japan), and the supernatant was transferred to a 1.5 mL tube. This LLE procedure was repeated twice to extract steroids from the remaining aqueous phase in high yield. The supernatant was dried by decompression centrifugation at 50 °C, and water with 20% methanol aqueous solution of 100 μL was added to the dried sample. The solution was injected into the LC/IM/MS.

To establish the matrix effect, the serum samples and 10-fold water-diluted serum samples were compared. The matrix effect was evaluated using the following formula: [peak area of steroid obtained by serum spiked with standard solution]/[(peak area of steroid obtained by 10-fold diluted serum spiked with standard solution) × 10].

### 4.3. Measurement Apparatus

LC/MS was performed using Acquity UPLC H-Class PLUS (Waters, Milford, MA, USA) and MALDI Synapt G2-Si HDMS (Waters) coupled with electrospray ionization (ESI) source. Data acquisition and processing were performed using MassLynx 4.2 and DriftScope 2.9 (Waters). The sample injection volume was 20 µL, and the column temperature was 42 °C. An Inertsil ODS-HL column (1.9 μm 1.0 × 100 mm, GL Sciences, Tokyo, Japan) was used at a flow rate of 0.1 mL/min. The mobile phase consisted of A (1 mM ammonium formate aqueous solution), B (acetonitrile), C (methanol), and D (THF and acetonitrile (*v*/*v*, 1/1)). Mobile phase D was used to clean the LC column after gradient separation. The gradient program occurred for 40 min as follows: 0 min to 0% B and 0% C, 1 min to 0% B and 0% C, 2.5 min to 18% B and 2% C, 13 min to 22.7% B and 2.3% C, 27 min to 58.5% B and 6.5% C, 31 min to 90% B and 10% C, 39 min to 90% B and 10% C, and 40 min to 100% D.

The ESI capillary voltage was set to 2.0 kV, and the sampling cone voltage was 30 V. The ion source and desolvation temperatures were 125 and 450 °C, respectively. The injection voltages into the trap and transfer cells were 4 and 2 V, respectively. Argon gas flowed into the trap and the transferred cells. The ion mobility cell traveling wave velocities increased from 650 to 1500 m/s with a wave height of 40 V. The trapping release time was 54 µs. N_2_ and helium gas were used in the helium and IMS cells, respectively. Sodium formate was used for the *m*/*z* calibration. Recalibration of the lock mass was performed using the fragmentation ion of leucine–enkephalin at *m*/*z* 397.187.

PCA was performed with [steroid peak area obtained from each subject]/[average of steroid peak area obtained from healthy subject 1 (*n* = 3)] using SIMCA-P+ ver. 12.0.1 software (Umetrics AB, Umea, Sweden). Each sample was measured in triplicate for PCA.

## 5. Conclusions

Twenty steroids in the serum were detected with high sensitivity and selectivity using LC/IM/MS. Although quantitative analysis via MS/MS is generally performed using multiple reaction monitoring, the application of ion mobility instead of the fragmentation method has better selectivity and an improved S/N ratio. In addition, this method can simultaneously perform non-targeted analysis of hydrophobic compounds, such as lipids, which are maintained in the same pretreatment as steroids. This analytical method is useful for establishing pathogenesis and elucidating the mechanism of aldosterone-producing adenomas, which are undergoing remarkable molecular developments, and for advancing medical research on glucocorticoids and sex steroids. For example, the ratio of precursor to product can be used to evaluate the activity of steroid enzymes. In addition, the ability of this method to identify non-targeted compounds, even if they are not commercially available and cannot be purchased as standards, would allow relative comparisons of the abundance of unknown compounds among subjects. This would be useful for finding new unknown biomarkers. Steroids and steroid-like compounds are also present in plants, mammals, and prokaryotes. The detection and quantification of steroids and steroid-like substances in these organisms could lead to a wide range of biological advances.

## Figures and Tables

**Figure 1 ijms-23-13858-f001:**
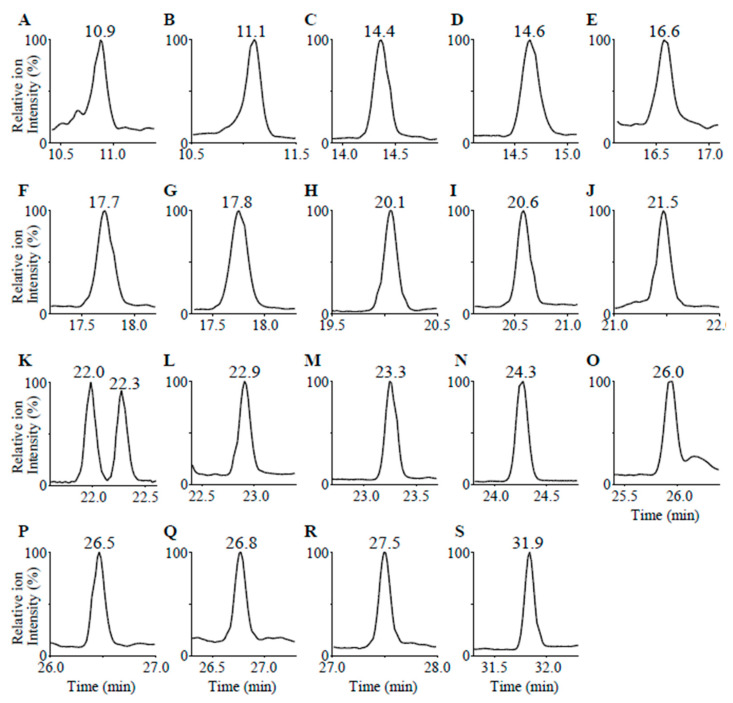
Extracted ion chromatograms of steroid standard solution with each specific DT at concentrations of 10 ng/dL. The *m*/*z* values and DT conditions of the compounds for the extracted ion chromatogram are as follows (*m*/*z* value, DT): (**A**) 18-oxocortisol at (377.197, 56), (**B**) 18-hydroxycortisol at (379.213, 57), (**C**) 18-hydroxycorticosterone at (363.217, 55), (**D**) aldosterone at (361.200, 54), (**E**) 19-hydroxy-4-androstene-3,17-dione at (303.196, 44), (**F**) cortisone at (361.203, 55), (**G**) cortisol at (363.218, 56), (**H**) 11-ketotestosterone at (303.196, 44), (**I**) 18-hydroxy-11-deoxy corticosterone at (347.222, 54), (**J**) adrenosterone at (301.183, 43), (**K**) 21-deoxycortisol at (347.223, 52) and corticosterone at (347.223, 54), (**L**) 11-deoxycortisol at (347.225, 55), (**M**) 16α-hydroxyprogesterone at (331.228, 52), (**N**) 11β-hydroxyprogesterone at (331.227, 50), (**O**) testosterone at (289.217, 43), (**P**) 11-deoxycorticosterone at (331.229, 53), (**Q**) androstenedione at (287.202, 43), (**R**) 17α-hydroxyprogesterone at (331.229, 51), and (**S**) progesterone at (315.233, 49).

**Figure 2 ijms-23-13858-f002:**
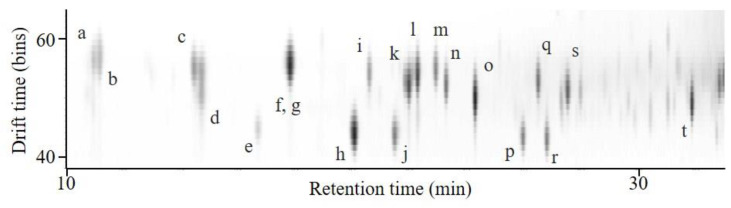
Two-dimensional plots of drift time–retention time (DT-RT) in the RT range of 10–33 min, DT 38–65 bins, and *m*/*z* 285–380. 18-oxoF (a), 18-OHF (b), 18-OHB (c), aldosterone (d), 19-OHA4 (e), cortisone (f), cortisol (g), 11-KT (h), 18-OH-DOC (i), adrenosterone (j), 21-deoxycortisol (k), corticosterone (l), 11-deoxycortisol (m), 16α-OHP (n), 11β-OHP (o), testosterone (p), DOC (q), androstenedione (r), 17α-OHP (s), and progesterone (t).

**Figure 3 ijms-23-13858-f003:**
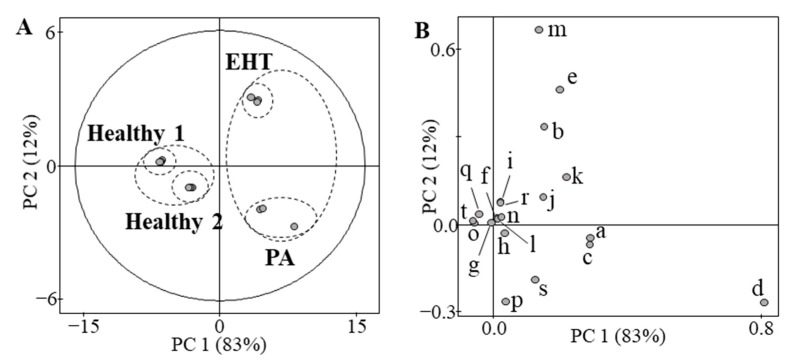
Principal component analysis (PCA) in the steroid profiling of four human serum samples. (**A**) Score plots. There are three plots in each serum. The plots model 20 steroids detected in LC/IM/MS of all sera. The X-axis and Y-axis describe the first and second principal components (PCs), respectively. Each sample was measured in triplicate for PCA. The PCAs were performed with the [steroid peak area obtained from each subject]/[average of steroid peak area obtained from healthy subject 1 (*n* = 3)]. These PCA plots were created using the relative value of steroids. PCA analysis shows that the distribution of steroid profiles differs significantly among individuals. (**B**) Loading plots. The PCA loading plots are provided for 20 steroids of 18-oxoF (a), 18-OHF (b), 18-OHB (c), aldosterone (d), 19-OHA4 (e), cortisone (f), cortisol (g), 11-KT (h), 18-OH-DOC (i), adrenosterone (j), 21-deoxycortisol (k), corticosterone (l), 11-deoxycortisol (m), 16α-OHP (n), 11β-OHP (o), testosterone (p), DOC (q), androstenedione (r), 17α-OHP (s), and progesterone (t).

**Figure 4 ijms-23-13858-f004:**
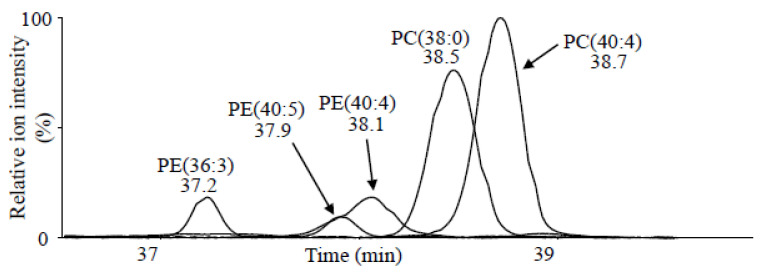
Overlaid extracted ion chromatograms via selection of *m*/*z* values of phosphatidylcholine (PC) (38:0), PC(40:4), phosphatidylethanolamine (PE) (36:3), PE(40:4), and PE(40:5) from a patient with PA in the range of RT 36.5–40.0 min using non-target LC/IM/MS.

**Table 1 ijms-23-13858-t001:** Summary of abbreviations, *m*/*z* value, retention time (RT), and drift time (DT) of 20 steroids obtained via LC/IM/MS. 18-oxocortisol (18-oxoF), 18-hydroxycortisol (18-OHF), 18-hydroxycorticosterone (18-OHB), 19-hydroxy-4-androstene-3, 17-dione (19-OHA4), 11-ketotestosterone (11-KT), 18-hydroxy-11-deoxy corticosterone (18-OH-DOC), 16α-hydroxyprogesterone (16α-OHP), 11β-hydroxyprogesterone (11β-OHP), 11-deoxycorticosterone (DOC), and 17α-hydroxyprogesterone (17α-OHP).

Compounds		Calculated Mass [M + H]^+^	Measured Mass [M + H]^+^	RT (min)	DT (bins)
a	18-oxoF	377.196	377.197	10.9	56
b	18-OHF	379.212	379.213	11.1	57
c	18-OHB	363.217	363.217	14.4	55
d	Aldosterone	361.201	361.200	14.6	54
e	19-OHA4	303.195	303.196	16.6	44
f	Cortisone	361.201	361.203	17.7	55
g	Cortisol	363.217	363.218	17.8	56
h	11-KT	303.195	303.196	20.1	44
i	18-OH-DOC	347.222	347.222	20.6	54
j	Adrenosterone	301.180	301.183	21.5	43
k	21-Deoxycortisol	347.222	347.223	22.0	52
l	Corticosterone	347.222	347.223	22.3	54
m	11-Deoxycortisol	347.222	347.225	22.9	55
n	16α-OHP	331.227	331.228	23.3	52
o	11β-OHP	331.227	331.227	24.3	50
p	Testosterone	289.216	289.217	26.0	43
q	DOC	331.227	331.229	26.5	53
r	Androstenedione	287.201	287.202	26.8	43
s	17α-OHP	331.227	331.229	27.5	51
t	Progesterone	315.232	315.233	31.8	49

**Table 2 ijms-23-13858-t002:** Internal standards (IS), IS concentration (IS conc.), limit of detection (LOD), limit of quantitation (LOQ), analytical recoveries, and matrix effect of steroids in serum samples. 18-hydroxycortisol-d4 (18-OHF-d4), aldosterone-d7, cortisone-d7, cortisol-d4, 11-ketotestosterone-d3 (11-KT-d3), 21-deoxycortisol-d8, corticosterone-d8, 11-deoxycortisol-d5, testosterone-d5, 11-deoxycorticosterone-d8 (DOC-d8), androstenedione-d3, 17α-hydroxyprogesterone-d8 (17α-OHP-d8), and progesterone-d9 were used as internal standards. The QC values of the recoveries were run in replicates (*n* = 5). The matrix effect was established using [peak area of steroid obtained from serum spiked with standard solution]/(10 × [peak area of steroid obtained from 10-fold diluted serum spiked with standard solution]). Cortisone, cortisol, corticosterone, and testosterone were evaluated using a 100-fold diluted serum. The concentrations of all steroids are expressed as ng/dL. LOD was obtained at an S/N ratio concentration of ≥3. The LOQ was obtained with a concentration of S/N ratio ≥5 and RSD <20%. The 100-fold water-diluted serum samples were prepared for analysis of cortisone, cortisol, corticosterone, and testosterone.

Compounds		IS	ISconc. (ng/dL)	LOD (ng/dL)	LOQ (ng/dL)	RSD of LOQ (%)	Recovery (%)	Matrix Effect
a	18-oxoF	18-OHF-d4	300	2.0	3.0	4.7	92	1.04
b	18-OHF	18-OHF-d4	300	0.5	1.0	7.5	95	1.18
c	18-OHB	Aldosterone-d7	400	1.0	1.0	9.2	115	1.01
d	Aldosterone	Aldosterone-d7	400	1.0	1.0	12.5	100	0.99
e	19-OHA4	Cortisone-d7	50	3.0	3.0	17.8	95	1.03
f	Cortisone	Cortisone-d7	50	1.0	1.0	8.2	105	0.99
g	Cortisol	Cortisol-d4	1000	0.5	1.0	17.4	86	1.16
h	11-KT	11-KT-d3	300	0.5	1.0	13.6	94	0.84
i	18-OH-DOC	11-KT-d3	300	1.0	1.0	12.6	102	1.12
j	Adrenosterone	21-Deoxycortisol-d8	300	0.5	1.0	12.8	94	0.77
k	21-Deoxycortisol	21-Deoxycortisol-d8	300	0.5	1.0	6.5	95	0.86
l	Corticosterone	Corticosterone-d8	100	0.5	1.0	4.8	95	1.04
m	11-Deoxycortisol	11-Deoxycortisol-d5	40	1.0	1.0	8.3	98	0.89
n	16α-OHP	11-Deoxycortisol-d5	40	0.5	1.0	9.1	98	0.83
o	11β-OHP	Testosterone-d5	50	0.5	1.0	8.1	87	0.68
p	Testosterone	Testosterone-d5	50	1.0	1.0	19.6	107	1.13
q	DOC	DOC-d8	25	0.5	1.0	14.6	103	0.65
R	Androstenedione	Androstenedione-d3	30	0.5	1.0	18.9	74	0.62
S	17α-OHP	17α-OHP-d8	25	0.5	1.0	18.0	116	0.67
T	Progesterone	Progesterone-d9	30	0.5	1.0	5.0	77	0.63

**Table 3 ijms-23-13858-t003:** Steroid profiling in four human serum samples. Serum samples were obtained from two healthy subjects: one with essential hypertension (EHT) and one with primary aldosteronism (PA). Units of all compounds are shown as ng/dL.

Compounds		Healthy 1 (ng/dL)	Healthy 2 (ng/dL)	EHT (ng/dL)	PA (ng/dL)
a	18-oxoF	3.3	4.6	11	12
b	18-OHF	35	116	178	98
c	18-OHB	11	45	36	41
d	Aldosterone	1.9	8.1	12	17
e	19-OHA4	5.6	7.1	30	14
f	Cortisone	1580	6030	4780	3550
g	Cortisol	9130	24,600	25,800	16,100
h	11-KT	17	24	20	19
i	18-OH-DOC	11	16	12	8.6
j	Adrenosterone	16	14	48	33
k	21-Deoxycortisol	1.4	<1	3.6	3.6
l	Corticosterone	140	480	450	260
m	11-Deoxycortisol	31	37	134	27
n	16α-OHP	26	16	20	22
o	11β-OHP	6.7	<1	<1	<1
p	Testosterone	532	1130	378	1240
q	DOC	8.6	4	3.4	1.8
r	Androstenedione	73	39	72	50
s	17α-OHP	143	106	115	176
t	Progesterone	119	4.1	2.5	2.9

## Data Availability

The de-identified participant data will be shared on a reasonable request basis. Data are available from Issey Osaka with permission from Toyama Prefectural University.

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
