# Peer review of "Comprehensive Steroid Assay with Non-Targeted Analysis Using Liquid Chromatography Ion Mobility Mass Spectrometry"

_ijms, 2022, doi:10.3390/ijms232213858_

Round 1

Reviewer 1 Report

M. Yamakawa et al. describe an analytical method LC/IM/MS for the steroid profiling of human serum samples. The scope and limitations of the methodology in the detection of aldosterone-related disorder have been clearly presented. Despite the small number of samples from healthy individuals and patients, at least 4 steroids have been identified as biomarkers related to the disease.

As the authors discuss, there are several issues that need to be addressed in the future. However the clear and detailed description of the already obtained results are particularly interesting and helpful for optimizing the method in order to discover reliable biomarkers.

I believe that the present form of the manuscript is good enough to be published in the International Journal of Molecular Sciences.

Author Response

Dear Reviewer 1

Thank you for taking the time to review our work.

We think the conventional LC MS/MS method is also a very useful method.

We believe that this new method will provide new insights for future steroidology.

We sincerely thank you for agreeing to publish our work in the International Journal of Molecular Sciences. We will continue to work very hard for the development of this research field.

Reviewer 2 Report

M. Yamakawa et al proposes a new methodology to measure twenty steroids in human serum.

The article, however, suffers from several defects that require a thorough revision before it can be published and make its acceptance unfeasible.

Lines 69-77: “… only a few qualitative and quantitative methods for steroid use using IM have been reported …”. They should be cited here, and the latter extended discussion (lines 238-258) must indicate the advantages of this particular methodology concerning the previously described methods.

Lines 137-141: two calibration models are proposed, but the one used in the paper must be indicated.

Line 144: the sentence “Serum samples contained both endogenous and spiked steroids” is incomplete.

Line 149: the sentence “The recovery rates of almost all steroids were sufficiently high” is a qualitative impression, but the results have to be tested against the values expected by an accepted biochemical method validation protocol.

Line 151: it is usual to calculate the LOQ as 10 times the S/N ratio. Why have you used a value of 5?

Table 2 / lines 171-177: low recoveries are obtained for five of the last six compounds of the table. They do not present a “small matrix effect and high recovery rate”.

Line 178: Four samples are an insufficient number to draw general conclusions, or even, to perform a PCA analysis.

Lines 182-190: there is a complete lack of information about how PCA has been made (variables employed, data pretreatment, software employed, …). With only four samples, a high variance explained by the two first PCs is expected (in this case, 93%). But it does not mean anything. Some values are below the LOD; how have you dealt with this fact? How have you selected the 15 steroids in the PCA model if you measure 20?

Figure 3: Why does it appear three/four points per sample in the score plot?

Lines 205-211. The lipids found at the end of the chromatogram, do they have any interest in the methodology?

4.2- Sample preparation. If I have to reproduce your methodology, I will find some steps hard to follow. Some examples:

line 336: “The 100-fold diluted serum samples were prepared for analysis of cortisone, cortisol, corticosterone, and testosterone.” How was it prepared?

Lines 344-345: “The supernatant was dried by decompression centrifugation at 50 °C, and water with 20% methanol was added to the dried sample.”  How much water?

Lines 347-350: Usually, the matrix effect is tested against water-prepared calibrations. If you do that, probably your matrix effect will increase more.

Lines 357-363: I can imagine that the separation starts with 100% A (but it is not clear because you also have solvent D). In addition, why are you using a mixture of acetonitrile and methanol? Does it provide a real advantage against the use of only acetonitrile?

Lines 364-371: some information about coupling IM and MS must be provided.

Supplementary material: do not show the calibration graphs (or not only). Present a table with all the statistics expected (intercept and its uncertainty, slope and its uncertainty, lineal interval, number of calibration points, LOD, LOQ, …)

Non-published material: it is written in Japanese, so it has limited utility. If it is an important material, translate it into English.

Author Response

Dear Reviewer 2.

Thank you for taking the time in your busy schedule to peer review our work.

Please see below as we will respond to your constructive comments one by one in good faith.

Comments 1: Lines 69-77: “… only a few qualitative and quantitative methods for steroid use using IM have been reported …”. They should be cited here, and the latter extended discussion (lines 238-258) must indicate the advantages of this particular methodology concerning the previously described methods.

Response 1: Thank you for pointing this out. We fully agree with your suggestion. We have added 2 new references and the sentence ‘’Hernandez-Mesa et al. evaluated the collision cross section (CCS) of steroid ions obtained by IM/MS to identify the steroid isomers [14]. Arthur et al. quantitatively analyzed various anabolic steroid metabolites in urine using nontarget LC/IM/MS analysis [15]. ‘’ to line 72-76. Furthermore, we have added the sentence ‘’ Hernandez-Mesa and Arthur separated steroid isomers by LC/IM/MS [14,15]. To improve the separation of isomers by IM, formed steroid alkali metal adduct ions [M+ alkali metal]+ were used instead of proton adduct ions [M+H]+.”  in lines 266-268.

  1. Hernandez-Mesa, M.; Bizec, B.L.; Monteau, F; García-Campañ, A.M.; Dervilly-Pinel, Collision Cross Section (CCS) Database: An Additional Measure to Characterize Steroids. Anal. Chem. 2018, 90, 4616−4625.
  2. Arthur K.L., Turner M.A., Brailsford A.D., Kicman A.T., Cowan D.A., Reynolds J.C., Creaser C.S. Rapid Analysis of Anabolic Steroid Metabolites in Urine by Combining Field Asymmetric Waveform Ion Mobility Spectrometry with Liquid Chromatography and Mass Spectrometry. Chem. 2017, 89, 7431–7437.

Comment 2: Lines 137-141: two calibration models are proposed, but the one used in the paper must be indicated.

Response 2: We apologize for the confusing explanation. We use only calibration models with first-order equations in this study. We changed the sentence from “The calibration curves for a wide range of concentrations yielded a quadratic curve because of the detector saturation. In contrast, the calibration curves of all steroids yielded good linearities for a range of low concentrations.’’ to “… they all yielded good linearities, …” in lines 141-142.

Comment 3: Line 144: the sentence “Serum samples contained both endogenous and spiked steroids” is incomplete.

Response 3: The sentence repeated the same meaning as the previous sentence. We have deleted the sentence “Serum samples contained both endogenous and spiked steroids”.

Comment 4: Line 149: the sentence “The recovery rates of almost all steroids were sufficiently high” is a qualitative impression, but the results must be tested against the values expected by an accepted biochemical method validation protocol.

Response 4: For general biochemical validation, a recovery rate of at least 80% is required. In this study, the recovery rates were achieved on almost steroids as shown in Table 2. We give the qualitative impression as you pointed out. We changed the sentence’’ The recovery rates of almost all steroids were sufficiently high.’’ to ‘’Recovery rates for most steroids were greater than 80%.’’ in lines 150-151.

Comment 5: Line 151: it is usual to calculate the LOQ as 10 times the S/N ratio. Why have you used a value of 5?

Response 5: A limit of quantitation of less than 20% RSD is recommended generally. In this experiment, it was achieved. On the other hand, the standard of S/N for LOQ is not clear. Recently, there are many studies based on S/N = 10 for the LOQ, but there are also reports on the use of S/N = 5 for LOQ. Research papers adopting S/N = 5 as the definition of LOQ are listed below. Therefore, we selected less than 20% RSD and S/N = 5 in this study.

・KateÅ™ina Hájková, Lukáš Mikulů, David Sýkora, Martin KuchaÅ™, Combination of UV and MS/MS detection for the LC analysis of cannabidiol-rich products, Talanta, 219(1) 2020, 121250

・Madina Tursumbayeva, Jacek A. Koziel, Devin L. Maurer, Bulat Kenessov, Somchai Rice, Development of Time-Weighted Average Sampling of Odorous Volatile Organic Compounds in Air with Solid-Phase Microextraction Fiber Housed inside a GC Glass Liner: Proof of Concept, Molecules, 24, 2019, 406

・Luis J Nùñez-Vergara, J. ASquella, J. CSturm, H Baez, Cristián Camargo, Simultaneous determination of melatonin and pyridoxine in tablets by gas chromatography-mass spectrometry, Journal of Pharmaceutical and Biomedical Analysis, 26(5-6), 2001, 929

Comment 6: Table 2 / lines 171-177: low recoveries are obtained for five of the last six compounds of the table. They do not present a “small matrix effect and high recovery rate”.

Response 6: We apologize for the confusing explanation. The matrix effect value of 1.00 gives the smallest effect. It means that there is a matrix effect both when it is greater than 1 and when it is less than 1. The two steroids gave poor recovery rates because matrix effect values were lower than 1. However, this text has been revised as it contains exceptions. We have added the sentence to the text as follows: “Androstenedione and progesterone with high matrix effect had low recovery rates.’’ in line 177-178. We have removed the sentence from the text as follows: ‘’The serum contains various hydrophobic lipids. It is plausible that a large amount of these lipids causes the matrix effect.’’.

Comment 7: Line 178: Four samples are an insufficient number to draw general conclusions, or even, to perform a PCA analysis.

Response 7: As clarified in the Introduction, the purpose of this study is to establish the LC IM/MS methodology. This study does not aim to conclude differences in general steroid profiling by hypertension status. We have shown that steroid profiles vary widely among individuals. To avoid any misunderstanding, we would like to add " PCA analysis shows that the distribution of steroid profiles differs significantly among individuals.” to the Legend of Figure 3. Thank you for your comment.

Comment 8: Lines 182-190: there is a complete lack of information about how PCA has been made (variables employed, data pretreatment, software employed, …). With only four samples, a high variance explained by the two first PCs is expected (in this case, 93%). But it does not mean anything. Some values are below the LOD; how have you dealt with this fact? How have you selected the 15 steroids in the PCA model if you measure 20?

Response 8: Thank you for your comment. PCA was performed with [steroid peak area obtained from each subject] / [average of steroid peak area obtained from healthy subject 1 (n = 3)] using SIMCA-P+ ver. 12.0.1 software (Umetrics AB, Umea, Sweden). The sentence has been added to the method section in the text. The first version of the figure used only 15 steroids because of the high steroid concentration requirement for PCA. However, we changed the representation to a PCA analysis using 20 different steroids, as this is confusing to the reader. We have changed Figure 3 and the legend. The purpose of this experiment is to show that PCA is possible with data obtained by LC/IM/MS. The objective was achieved in this experiment.

Comment 9: Figure 3: Why does it appear three/four points per sample in the score plot?

Response 9: The number of plots in the score plots is three each.

Comment 10: Lines 205-211. The lipids found at the end of the chromatogram, do they have any interest in the methodology?

Response 10: This LC/IM/MS method enables both high-sensitivity quantitative analysis of steroids as targeted analysis and non-targeted analysis. It showed potential as a non-targeted analysis to detecting other metabolite lipids.

Comment 11: 4.2- Sample preparation. If I have to reproduce your methodology, I will find some steps hard to follow. Some examples: line 336: “The 100-fold diluted serum samples were prepared for analysis of cortisone, cortisol, corticosterone, and testosterone.” How was it prepared?

Response 11: Thank you for your comment. Our explanation was inadequate. Serum was diluted 100-fold in water and spiked with internal standard solution for quantitation analysis of cortisone, cortisol, corticosterone, and testosterone. The solution was prepared in the same method as other sera, and was measure using LC/IM/MS. We have changed the sentence ‘‘The 100-fold diluted serum samples were prepared for analysis of cortisone, cortisol, corticosterone, and testosterone.’’ to ‘‘The 100-fold water-diluted serum samples were prepared for analysis of cortisone, cortisol, corticosterone, and testosterone.’’ in line 349-350 and figure legend of Table 2.

Comment 12: Lines 344-345: “The supernatant was dried by decompression centrifugation at 50 °C, and water with 20% methanol was added to the dried sample.” How much water?

Response 12: Thank you for your comment. Our explanation was inadequate. We have changed the sentence ‘’The supernatant was dried by decompression centrifugation at 50 °C, and water with 20% methanol was added to the dried sample’’ to ‘’The supernatant was dried by decompression centrifugation at 50 °C, and water with 20% methanol aqueous solution of 100 μL was added to the dried sample.’’ in lines 357-359.

Comment 13: Lines 347-350: Usually, the matrix effect is tested against water-prepared calibrations. If you do that, probably your matrix effect will increase more.

Response 13: Thank you for your comment. There are three ways to evaluate the matrix effect. 1: Comparison of blank water and standard-spiked serum extraction, 2: comparison of a surrogate matrix solution and standard-spiked serum extraction, 3: evaluation using a diluted serum. Method 3 can evaluate the matrix effect for each actual sample in detail. Therefore, method 3 was used to evaluate the matrix effect in this study. The purpose of the experiment was a sensitive quantitative analysis, which was achieved under conditions including matrix effects. Therefore, it is considered that the matrix effect did not have a large influence.

Comment 14: Lines 357-363: I can imagine that the separation starts with 100% A (but it is not clear because you also have solvent D). In addition, why are you using a mixture of acetonitrile and methanol? Does it provide a real advantage against the use of only acetonitrile?

Response 14: In the conventional quantitative analysis of steroids in serum, continuous measurement of samples is difficult due to deterioration of the column. It was necessary to wash the column. The method is not described in the previous paper. The few reports that consider cleaning are listed below.

・Sophia Rehm, Katharina M. Rentsch, LC-MS/MS method for nine different antibiotics, Clinica Chimica Acta, Clinica Chimica Acta 511 (2020) 360–367

In this study, washing the column using mobile phase D enabled continuous measurement of serum extraction samples. The Acetonitrile mobile phase in the separation of steroid LC/MS causes steroids to accumulate on the LC column. We used methanol which could dissolve steroids to suppress it. Acetonitrile was used to reduce the pressure in the LC system.

Comment 15: Lines 364-371: some information about coupling IM and MS must be provided.

Response 15: We have provided sufficient setting conditions information based on previous IM/MS reports. We showed the system in graphical abstract that the reader first focuses on in order not to mislead.

Comment 16: Supplementary material: do not show the calibration graphs (or not only). Present a table with all the statistics expected (intercept and its uncertainty, slope and its uncertainty, lineal interval, number of calibration points, LOD, LOQ, …)

Response 16: We show the LOD and LOQ in Table 2. Information of slope was added in Supplementary Figure 1. We have added the concentration information of steroids for creating calibration curves in the text for an explanation of lineal interval. We have revised the sentence ‘’The serum sample (100 µL) was mixed with 5 µL methanol and 5 µL internal standard solution in a 1.5mL tube.’’ to ‘’Sera containing 1, 2, 3, 10, 30, 100, and 300 ng/dL steroids were prepared by the addition of 5 µL methanol solution of steroid standard. The solution was mixed with 5 µL internal standard solution in a 1.5 mL tube. These solutions were used for creation of calibration curves. ‘’ in lines 346-349.

Comment 17: Non-published material: it is written in Japanese, so it has limited utility. If it is an important material, translate it into English.

Response 17: A translation from Japanese to English is transcribed below.

Case Report Consent Form

Before presenting my case report at domestic and international conferences and research meetings and submitting it to academic journals in Japanese and English, I provided sufficient time for the patient (the surrogate) to receive an explanation based on the consent document and to decide whether or not to present it as a "case report" or to prepare and submit a paper on the case.

[item explained].

  1. Purpose
  2. How to publish
  3. Freedom to Cooperate and Cancel
  4. Protection of Human Rights and Personal Information

In principle, strict measures should be taken to protect personal information, and information that could lead to the identification of individuals should not be included. Notwithstanding the aforementioned considerations, if there is a possibility that an individual may be identified, consent should be obtained from the patient or a surrogate.

  1. Management of the name and contact information of the person responsible for the "case report" (first author or responsible author) and the consent form.

Explainer                           

(The first or responsible author is preferred, but other authors are acceptable.)

Letter of intent

I have received an explanation based on the consent document from the above-mentioned person explaining that I will prepare and submit an article as a "case report" for an English-language academic journal, and I fully understand and accept the contents of the explanation.

I agree that the person responsible for the "case report" may make presentations at conferences, etc. and prepare and submit papers.

Date of Consent:                             

Signature:                                  

Signature of Agent:                 (relationship)

Description of Consent Form for "Case Reporting”

  1. Purpose

The purpose is to contribute to the development of medical diagnosis and treatment by sharing with medical professionals and others the details of cases considered important for the practice of medicine.

  1. How to publish

The content of the presentation will be made public at the conference and in the abstract book.

 Research results will be available in journals and on life science search sites such as PubMed.

  1. Freedom to Cooperate and Cancel

(1) The patient (or the agent) will decide whether or not to present the case or to prepare and submit an article.

(ii) Refusal to make a presentation or prepare and submit an article will not cause any disadvantage in medical treatment or otherwise.

(iii) It is also possible to withdraw consent even after consent has been given.

(iv) If there is a request for withdrawal of a paper after publishment, the editorial board of the journal will discuss the request and consider how to respond to it.

  1. Protection of Human Rights and Personal Information

The best care should be taken to protect human rights and personal information, and no information related to patient privacy should be included, except for items essential to the presentation. Specifically, the following information should not be included: name, date of birth, date of medical examination (date), place of birth or residence, information that identifies family or lineage, photographs that identify individuals, and other information that leads to identification of individuals should not be included. If, despite the above considerations, there is a possibility that an individual may be identified, the consent of the patient or a surrogate should be obtained.

  1. Name and contact information of the person responsible and management of the consent form

The explainer should provide the name and contact information of the person responsible. The first author or responsible author is responsible for managing and storing the consent form and ensuring that leakage and loss of personal information are prevented. The patient/substitute author(s) should retain a copy of the consent form as well as a copy of the explanation in the consent form.

Reviewer 3 Report

This article describes a comprehensive analysis of steroids required for the study of mobile ions in serum by liquid chromatography and mass spectrometry. The authors created a unique highly sensitive analysis system by dividing 20 steroids based on their retention time, m/z value and drift time. These twenty steroids were measured in the blood serum of several patients with primary aldosteronism, essential hypertension and healthy subjects and were clearly classified using the analysis of the main components. This analytical method has the potential to elucidate deeper features of the pathogenesis of APA and identify unknown steroids that could serve as biomarkers for APA with various genetic mutations. The article is written in an accessible and understandable language. All the necessary supporting proofs have been completed. Thus, after a minor revision, I could recommend this article for publication in the International Journal of Molecular Sciences.

1. The conclusions need to be expanded on the details of some of the previously given values obtained using LC/IM/MS.
2. What is the basis for choosing exactly these 20 steroids? What is the general rationale for this regarding the presence of various functional groups and fused heterocycles. What is the pattern? Give structural formulas as examples.
3. Please give another figure 2 in the best quality.

Author Response

Dear Reviewer 3

Thank you for taking the time to review our work.

We agree with you that your comments are important for readers to understand the content and potential of this study.

We will respond to each of your comments below.

We strongly believe that your constructive comments have added value to our study.

We sincerely appreciate your dedication to science.

Comments 1. The conclusions need to be expanded on the details of some of the previously given values obtained using LC/IM/MS.

Response 1.

We agree with your suggestion.

We added the text " The activity of CYP11B2, an aldosterone synthase, is expressed as 18-oxocortisol (18-oxoF)/cortisol ratio or aldosterone/corticosterone ratio. The 18-oxoF/cortisol ratio was 0.000361, 0.000187, 0.000426, and 0.000745 for healthy subject 1, healthy subject 2, EHT patients, and PA patients, respectively. The aldosterone/corticosterone ratio was 0.013571, 0.016875, 0.026667, and 0.065385, respectively. Compared with that in the other subjects, CYP11B2 activity was the highest in patients with PA." to Line 182-188 of Result.

We added the text " For example, the ratio of precursor to product can be used to evaluate the activity of steroid enzymes. In addition, the ability of this method to identify non-targeted compounds, even if they are not commercially available and cannot be purchased as standards, would allow relative comparisons of the abundance of unknown compounds among subjects. This would be useful for finding new unknown biomarkers." to Line 397-402 of Conclusion.

This helped us to emphasize the usefulness of this analysis method to the reader. Thank you very much.

Comments 2. What is the basis for choosing exactly these 20 steroids? What is the general rationale for this regarding the presence of various functional groups and fused heterocycles. What is the pattern? Give structural formulas as examples.

Response 2. Our goal was to measure as many steroids as possible with LC IM/MS.

We collected available standards from commercial companies and initially, in addition to these 20 steroids, the following 7 were also candidates for measurement.

・beta-Estradiol 3,17-disulfate dipotassium salt

・(R,S)-Equol

・Pregnenolone sulfate sodium salt

・5-androstene-3beta,16alpha,17alpha-triol

・5alpha-dihydro-11-keto Testosterone

・beta-Estradiol

・Estrone

However, these standards were not included in this study because they were not detected with high sensitivity in positive ion mode.

Acidic OH group in estradiol and estrone and equol and sulfate group in sulfate conjugate of steroids prevents the formation of [M+H]+ in positive ion mode in the LC/IM/MS. On the other hand, these steroids could be detected in negative ion mode sensitively as [M-H]- ions. It was clear from our investigation that these have properties that are more easily detected in negative ion mode (unpublished data). In the future, we would like to report on a steroid measurement system that can be comprehensively detected in negative ion mode.

Comment 3. Please give another figure 2 in the best quality.

Response 3. We have modified the image by re-rendering it to improve the image quality.

However, the image quality of the raw data after drawing with DriftScope 2.9 (Waters) is poor to begin with, and we have enlarged the image to make it easier for the reader to see the chromatographic peaks. Further improvement of image quality is difficult.

There may be some areas of blurriness, but we do not believe that the quality of the image is such that it will mislead the reader.

Round 2

Reviewer 2 Report

M. Yamakawa et al. propose a new methodology to measure twenty steroids in human serum.

The revised article has improved a little bit in the redaction and exposition. But no changes have been made in the number of samples analyzed. Trying to obtain the presented conclusions is clearly not supported by the (insufficient) experimental data. The distances between the two sane patients and the two with diseases are the same, so you cannot decide anything about the patient’s health state as it is claimed as a big hit in the paper.

In addition, in the supplementary material, the statistics expected for the calibration lines are still missing: intercept and its uncertainty, slope and its uncertainty, lineal interval, number of calibration points, etc.

Only with a significant number of more samples, the paper can be revised to check if the conclusions presented are correct.